# β-Cyclodextrin Inhibits Monocytic Adhesion to Endothelial Cells through Nitric Oxide-Mediated Depletion of Cell Adhesion Molecules

**DOI:** 10.3390/molecules25163575

**Published:** 2020-08-06

**Authors:** Sujeong Jang, Seongsoo Lee, Heonyong Park

**Affiliations:** Department of Molecular Biology & Institute of Nanosensor and Biotechnology, Dankook University, 119 Dandae-ro, Dongnam-gu, Cheonan-si, Chungnam 31116, Korea; 72191414@dankook.ac.kr (S.J.); 32153223@dankook.ac.kr (S.L.)

**Keywords:** cyclodextrin, nitric oxide synthetase, protein kinase C-epsilon, cell adhesion molecule, THP-1 cell adhesion

## Abstract

Cyclodextrins (CDs) are used as drug delivery agents. In this study, we examined whether CDs have an inflammatory effect on endothelial cells. First, we found that β-CD promoted cell proliferation in bovine aortic endothelial cells and elevated nitric oxide (NO) production through dephosphorylation of threonine-495 (T-495) in endothelial nitric oxide synthetase (eNOS). Dephosphorylation of T-495 is known to activate eNOS. Phosphorylation of T-495 was found to be catalyzed by protein kinase Cε (PKCε). We then found that β-CD inhibits binding of PKCε to diacylglycerol (DAG) via formation of a β-CD-DAG complex, indicating that β-CD inactivates PKCε. Furthermore, β-CD controls activation of PKCε by reducing the recruitment of PKCε into the plasma membrane. Finally, β-CD inhibits expression of intercellular and vascular cell adhesion molecule-1 by increasing NO via control of PKCε/eNOS and suppression of THP-1 cell adhesion to endothelial cells. These findings imply that β-CD plays an important role in anti-inflammatory processes.

## 1. Introduction

Cyclodextrins (CDs) consist of several glucopyranoses joined by an α-(1,4)-glycosidic linkage, forming a cyclic truncated cone-like structure with a hydrophilic outer surface and a hydrophobic inner cavity [1,2]. Six-, seven-, and eight-membered CDs are called α-, β-, and γ-CDs, respectively. In the 1970s, CDs were biotechnologically produced from starch by catalysis using cyclodextrin glycosyltransferase [3]. Currently, the production of pure α-, β-, and γ-CDs is well-established, and the compounds are commercially available [4].

CDs are utilized as drug delivery vehicles because they contain a hydrophobic cavity, into which hydrophobic organic compounds can be incorporated. The formation of CD complexes with diverse hydrophobic guest compounds alters their physicochemical properties by, for example, increasing wettability [1]. Accordingly, CDs are widely utilized as delivery agents in a variety of industrial areas; they molecularly encapsulate guest molecules in the solid state or in solution. Over the last two decades, CDs have been chemically and biochemically modified to improve efficacy. For example, methyl-β-CD is chemically produced by methylation of the hydroxyl groups of β-CD. Methyl-β-CD extracts cholesterol from the plasma membrane and depletes lipid rafts [5,6]. Thus, a broad spectrum of modified CDs has been developed and utilized for drug delivery systems. 

Recently, evidence has accumulated regarding the biomedical applications of β-CD, which forms complexes with various anti-inflammatory drugs [7]. The efficacy of lipophilic anti-inflammatory agents, such as loteprednol etabonate, piroxicam, valecoxib, and p-cymene, is enhanced by formation of an inclusion complex in CDs [8,9,10,11]. For example, rat hind paw edema inhibition is elevated 5–6-fold in the valecoxib-CD complex, compared to a single dose of valecoxib [10]. Moreover, the topical anti-inflammatory effect of piroxicam is maintained for longer periods by formation of an inclusion complex in CDs [9]. Collectively, CDs enhance the stability, anti-inflammatory efficacy, and bioavailability of various anti-inflammatory drugs through formation of inclusion complexes. 

Although CDs are widely utilized in a variety of biomedical and industrial applications, the pro- or anti-inflammatory functions of CDs themselves have not yet been studied. Inflammation is an immune response to infection or xenobiotic toxins. In the early stages of inflammation, immune cells including leukocytes adhere to the endothelium of blood vessels near inflamed regions [12]. Leukocytic adhesion to endothelial cells is directly exerted through molecular interaction of cell adhesion molecules (CAMs). Intercellular cell adhesion molecule-1 (ICAM-1) and vascular cell adhesion molecule-1 (VCAM-1) are well-known CAMs that are responsible for heterotypic cell adhesion and are expressed in endothelial cells [13]. Therefore, elevated expression of ICAM-1 and VCAM-1 is crucial for the progress of inflammatory responses. Consistently, inflammatory stimuli including lipopolysaccharide (LPS) or tumor necrosis factor α (TNFα) promote expression of ICAM-1 and VCAM-1 through NFκB activation [14]. Meanwhile, inflammation triggers the development of various pathological conditions, such as atherosclerosis, ischemia, and osteoathritis [15]. In this respect, anti-inflammatory agents are medically beneficial for preventing the progress of inflammatory diseases.

Nitric oxide (NO) is the best-known gaseous molecule that dilates blood vessels [16]. Additionally, NO regulates inflammatory responses by modulating the expression of pro-inflammatory genes such as *icam-1* and *vcam-1* [17]. Altered NO production is strongly associated with inflammation in the vascular system. In this study, we determined that one of the CDs, β-CD, promotes NO expression and upregulates other anti-inflammatory responses in endothelial cells. The results of this study demonstrate the mechanisms of action (MoA) of CDs in inflammation and will provide further insight into the pharmacological effect of CDs on the vascular system.

## 2. Results

### 2.1. β-Cyclodextrin Promotes Cell Proliferation in Bovine Aortic Endothelial Cells

We first examined the cell proliferative effect of CDs at various concentrations. As displayed in Figure 1A,B, β-CD (400 μg/mL) promoted cell proliferation in bovine aortic endothelial cells (BAECs) by ~200%, whereas the increase in proliferation induced by α-CD and γ-CD was ≤ 40%. These results indicate that α-, β-, and γ-CDs are not cytotoxic to endothelial cells, confirming that these CDs can be safely used as drug delivery compounds. Moreover, β-CD is known to induce cell proliferation by unknown mechanisms.

### 2.2. β-Cyclodextrin Activates Endothelial Nitric Oxide Synthetase in Endothelial Cells

To determine how β-CD induces cell proliferation, we evaluated whether β-CD affects signaling molecules that play critical roles in endothelial cell proliferation. First, we found that β-CD did not affect ERK and Akt phosphorylation (Figure 2A,B). Then we tested whether β-CD had an effect on the activation of eNOS. As shown in Figure 2C, basal phosphorylation of threonine 495 (T495) was maximally reduced by 40% in BAECs treated with β-CD, whereas phosphorylation at serine 633 (S633) and 1177 (S1177) was not altered. It is well-established that phosphorylation at T495 of eNOS inhibits eNOS, whereas phosphorylation of S633 and S1177 activates eNOS [18,19,20]. Therefore, we assumed that β-CD elevated NO production through dephosphorylation of T495, leading to eNOS activation. Finally, we found that β-CD elevated NO production. As shown in Figure 2D,E, NO production was maximally increased by ~150% when BAECs were treated with β-CD. Furthermore, we confirmed that NO promoted cell proliferation in BAECs, which is consistent with a previous report [21].

### 2.3. Protein Kinase Cε Is a Key Regulator of eNOS T495 Phosphorylation in Bovine Aortic Endothelial Cells 

We next explored the upstream factors responsible for the eNOS dephosphorylation induced by β-CD. Caveolin-1 and intracellular calcium ions (Ca^2+^) are well-known upstream regulators of eNOS phosphorylation [22]. Therefore, we tested whether β-CD altered caveolin-1 or Ca^2+^ concentration. As shown in Figure 3A, β-CD had no effect on either the expression or phosphorylation of caveolin-1, indicating that caveolin-1 was not involved in β-CD-associated eNOS dephosphorylation. More convincingly, β-CD appeared to rarely disintegrate the caveolar structure (Appendix A). In addition, β-CD did not alter intracellular Ca^2+^ concentration (Figure 3B), indicating that Ca^2+^ concentration has no effect on β-CD-induced eNOS dephosphorylation. We further tested if PKCs, known upstream regulators of eNOS [23,24], were associated with T495 phosphorylation. PKC activators, such as phorbol 12-myristate 13-acetate (PMA) and diacylglycerol (DAG), increase T495 phosphorylation (Figure 3C,D), indicating that at least one PKC isozyme may be involved in T495 phosphorylation. Using a variety of PKC inhibitors, we examined which PKC isozyme was responsible for T495 phosphorylation. We found that only the PKCε inhibitor decreased T495 phosphorylation, whereas inhibitors of the other PKC isotypes had no effect on NO production or T495 phosphorylation (Figure 3E,F). When ectopic PKCε was expressed in BAECs, T495 phosphorylation was enhanced but NO production was decreased (Figure 3G). In contrast, T495 phosphorylation was significantly reduced [25], and NO production was promoted in PKCε-knockdown BAECs, compared to cells transfected with scrambled RNA (Figure 3H). These results confirmed that PKCε is a key enzyme in eNOS T495 phosphorylation. 

### 2.4. β-Cyclodextrin Alters Membrane Recruitment of Protein Kinase Cε, Thereby Prohibiting the Accessibility of Protein Kinase Cε to eNOS

β-CD was found to reduce PMA-promoted T495 phosphorylation and reverse the inhibitory effect of PMA on NO production (Figure 4A,B). These results imply that β-CD inhibits PKCε activation induced by DAG, a physiological analog of PMA. Consistently, β-CD was shown to inhibit DAG-induced eNOS T495 phosphorylation (Figure 4C). Furthermore, we found that the DAG-inhibitory effect of β-CD was exerted in the presence of fetal bovine serum (Figure 4D), suggesting that this inhibitory effect may occur in the vascular system in vivo. Next, we examined whether β-CD inhibits the binding between PKCε and DAG to determine how β-CD controls DAG-associated PKCε activation. For the in vitro binding assay, we obtained a purified GST-PKCε fusion protein (Appendix A). The in vitro binding results show that PKCε binds to DAG with a K_d_ of 74 ± 8 μM (Figure 4E). However, the binding affinity of PKCε against DAG was remarkably reduced by β-CD, demonstrated by an increase in the K_d_ of DAG to 486 ± 295 μM when DAG was pre-incubated with β-CD. This finding indicates that β-CD inhibits PKCε activation through molecular encapsulation of DAG. It remains unknown how β-CD in extracellular fluid affects intracellular PKCε. One report found that alterations of PKCε localization regulate downstream signaling [24]. Therefore, we tested whether β-CD modifies PKCε localization. Under fluorescence microscopy, we observed that ectopic GFP-PKCε was densely punctate, but rarely co-localized with caveolin-1 (Appendix A). These punctate fluorescent spots originating from GFP-PKCε disappeared with β-CD treatment (Figure 4F). This result indicates that β-CD induces delocalization of PKCε to cytosolic regions.

### 2.5. β-Cyclodextrin Inhibits THP-1 Cell Adhesion to Bovine Aortic Endothelial Cells by NO-Mediated Suppression of Cell Adhesaion Molecules

Monocytic adhesion to the endothelium, an early event in the inflammatory response, triggers the development of inflammatory diseases including atherosclerosis. To determine the inflammatory function of β-CD, we examined whether β-CD alters TNFα-induced monocytic adhesion to endothelial cells. In this study, we found that β-CD inhibited THP-1 cell adhesion to BAECs 1.6-fold, when compared to TNFα-promoted THP-1 cell adhesion (Figure 5A). This finding indicates that β-CD has an anti-inflammatory function in the vascular system. Furthermore, an eNOS inhibitor, L-NAME, reversed the inhibitory effect of β-CD. This result indicates that NO elevated by β-CD blocks THP-1 cell adhesion. In addition, real-time PCR analysis showed that β-CD inhibited expression of *icam-1* and *vcam-1* at the transcriptional level (Figure 5B). This inhibitory effect occurs via NO elevated by β-CD. Further mechanistic study determined that β-CD markedly inhibits TNFα-induced degradation of IκB (Figure 5C). Provided that IκB degradation leads to NFκB activation, β-CD suppresses *icam-1* and *vcam-1* expression through NFκB inactivation. As a result, β-CD increases NO, which blocks expression of *icam-1* and *vcam-1*, thereby inhibiting monocytic adhesion to endothelial cells. These findings provide insight into the anti-atherogenic function of β-CD.

## 3. Discussion

CDs were developed with the aim of efficiently delivering bioactive chemicals or drugs. Therefore, we examined if CDs themselves have any inflammatory function in the vascular system. First, we found that β-CD has cell proliferative effects in BAECs. Although controversial, the proliferation of endothelial cells is beneficial for vascular stability. Since NO-mediated proliferation of endothelial cells is of great benefit in wound healing [26], β-CD may be useful as a vasoactive compound for wound healing. This finding implies that β-CD can be utilized as a bioactive compound. Second, we found that β-CD promotes NO production in endothelial cells (Figure 2). Endothelial NO exerts anti-inflammatory activity [27], which provides an insight into the bioavailability of β-CD for the treatment and prevention of inflammatory diseases including atherosclerosis. Consistently, 2-hydroxypropyl-β-CD was reported to exert anti-inflammatory activity in an atherosclerotic mouse model [28]. It is established that enhanced NO increases cell proliferation in endothelial cells [21], and NO is a key mediator promoting the functionality of β-CD in endothelial cells. Here, we identified the detailed mechanisms of action responsible for the anti-inflammatory activity of unmodified β-CD at the molecular and cellular level. β-CD inhibits monocytic adhesion to endothelium via NO elevation, NFκB inactivation, and ICAM-1 or VCAM-1 depletion. These results highlight the novel bioavailability of β-CD for preventing inflammation and cardiovascular disease. Collectively, these results suggest that unmodified β-CD can be utilized as a bioactive material in a variety of areas, such as biomedicine, cosmetics, and bio-foods. 

The functionality of β-CD is attributable to the formation of CD complexes that include various lipid-soluble molecules, including DAG. We found that β-CD forms complexes with DAG and cholesterol. These guest molecules are membrane components, indicating that β-CD triggers the efflux of membrane components. Accordingly, we assume that extracellular β-CD alters the integrity of the plasma membrane. Alterations in the microenvironment in the plasma membrane might further affect downstream intracellular signaling cascades. The signaling molecule PKCε is crucial for β-CD-induced NO production. Unlike conventional PKCs, PKCε does not have a Ca^2+^ binding domain [24]. Domain-mapping supports our findings showing that the vascular function of β-CD is independent of intracellular Ca^2+^ concentration (see Figure 3B). Although the vascular functionality of β-CD is associated with the physicochemical properties of the plasma membrane, which are tightly linked to Ca^2+^ concentration, Ca^2+^-related signaling cascades are not involved in β-CD-induced NO production.

Our findings raise the question of how β-CD specifically blocks PKCε but not ERK and Akt. This issue is closely related to the behavior of β-CD outside the cell. β-CD cannot directly contact signaling molecules. It is more likely that β-CD alters the structure of lipid rafts or caveolae. Consistently, the active form of PKCε is found in lipid rafts (detergent-insoluble regions of the membrane) [24], the main component of which is cholesterol. β-CD binds to cholesterol, indicating that β-CD alters the physicochemical behavior of cholesterol in the plasma membrane. It is well-established that methyl-β-CD induces cholesterol efflux from the plasma membrane [29]. The efflux of cholesterol exerts cytotoxic effects. However, unmodified β-CD, even at 800 μg/mL (the highest concentration examined), did not induce cell death (Figure 1), suggesting that β-CD may not drastically provoke cholesterol efflux and cause complete disintegration of lipid rafts or caveolae. Furthermore, the present study suggested that β-CD prevents the accessibility of DAG to PKCε, as shown in Figure 4E. Given that DAG is both a membrane component and an effector of PKCε, β-CD’s interference with the binding between PKCε and DAG implies that β-CD provoked efflux of DAG from the plasma membrane, thereby preventing recruitment of PKCε to the plasma membrane. Accordingly, β-CD-induced efflux of DAG blocks PKCε-catalyzed phosphorylation of eNOS at T-495 owing to spatial segmentation between PKCε and eNOS, without causing large-scale disintegration of lipid rafts or caveolae (see Appendix A). Our findings suggest that unmodified β-CD subtly controls macro-signaling complexes in lipid rafts or caveolae. A working model of the anti-inflammatory effects of β-CD is shown in Figure 6. We further tested if β-CD has an anti-inflammatory effect on THP-1 cells. Interestingly, TNFα-induced THP-1 cell adhesion to endothelial cells was inhibited by treatment of THP-1 cells with β-CD (Appendix A). This result implies that β-CD has anti-inflammatory functions in a variety of cell lines. However, the detailed mechanisms by which β-CD exerts its anti-inflammatory functions in THP-1 cells remain to be elucidated. Finally, the results of the present study reveal the detailed mechanism of action of the anti-inflammatory effects of β-CD and newly suggest that β-CD is bio-available in the cardiovascular system.

## 4. Materials and Methods

### 4.1. Cell Culture, Reagents, and Treatments

Bovine aortic endothelial cells (BAECs) derived from descending thoracic aortas were cultured in Dulbecco’s Modified Eagle’s Medium (DMEM, Welgene Inc., Gyeongsan, Korea) supplemented with 20% fetal bovine serum (FBS, Welgene Inc.) and antibiotics (penicillin and streptomycin). For serum starvation, BAECs were incubated with DMEM containing 0.5% FBS for at least 2 h. The cells were grown in a humidified incubator in a 5% CO_2_ atmosphere at 37 °C. The cells used in this study were passaged 7 to 15 times. Different concentrations of α-, β-, and γ-CD were used to obtain dose–response curves. We used 400 μg/mL α-, β-, and γ-CD in the serum starvation medium for the experiments unless otherwise indicated. We also used various activators and inhibitors of PKCε. The activators used were phorbol 12-myristate 13-acetate (PMA, 100 nmol/L, Enzo Life Sciences Inc., Farmingdale, NY, USA) and glycerol-1,3-dipalmitate (diacylglycerol (DAG), 100 μmol/L, Sigma-Aldrich, St. Louis, MO, USA). The inhibitors included DHS (DL-dihydrosphingosine, Sigma) and GO6983 (Sigma) and a PKCε inhibitor peptide (20 μM, Santa Cruz, Dallas, TX, USA). The activators were utilized for time course experiments (0–30 min), and the inhibitors were utilized for pretreatment before β-CD treatment in the time-course experiments (0–60 min). We also used TNFα (ProSpec-Tany TechnoGene Ltd., Ness-Ziona, Isael) and L-NAME (Sigma) as a pro-inflammatory agent and an eNOS inhibitor, respectively.

### 4.2. Cell Viability Test

Cell viability was measured using a cell viability test kit (Daeil Lab Service Co., Seoul, Korea). Confluent BAECs were serum-starved for 2 h and then treated with various concentrations of α-, β-, and γ-CD for the indicated times. The cells were then reacted with 10-fold diluted WST-1 reagent (Daeil Lab Service Co.) for 2 h. The absorbance of the live cells was measured at 450 nm.

### 4.3. Western Blotting

Western blotting was executed as previously described [13]. Briefly, cell lysates were blotted with antibodies specific to p-ERK (Cell Signaling, Danvers, MA, USA), ERK (Cell Signaling), p-Akt (Cell Signaling), Akt (Cell Signaling), endothelial nitric oxide synthetase (eNOS, Cell Signaling), p-Ser1177 eNOS (Cell Signaling), p-Ser633 eNOS (Bioss Antibodies Inc., Woburn, MA, USA), p-Thr495 eNOS (Cell Signaling), protein kinase Cε (PKCε, Santa Cruz), p-Cav-1 (BD Biosciences, Franklin Lakes, NJ, USA), Cav-1 (BD Biosciences), IκB (Sigma), and β-actin (Santa Cruz). 

### 4.4. Measurement of Nitric Oxide

Confluent BAECs were serum-starved for more than 2 h and then treated with or without β-CD, PKCε activator, and various inhibitors for the indicated times. Subsequently, the cells were thoroughly washed with warm HEPES buffer (5 mM HEPES, 5 mM KCl, 1 mM MgCl_2_, 5 mM glucose, 140 mM NaCl, and 2 mM CaCl_2_, pH 7.4) and then treated with 1 μM A23187 (calcium ionophore, Sigma) for 20 min. Then the cells were treated with 10 μM DAF2-DA (Merck Millipore) for 30 min and harvested with HEPES buffer. Finally, the relative amounts of nitric oxide (NO) were measured by fluorescence intensity (excitation at 485 nm and emission at 515 nm). 

### 4.5. Calcium Assay

Confluent serum-starved BAECs were treated with β-CD for the indicated times. Then the cells were washed with HHBS buffer (1.26 mM CaCl_2_, 0.49 mM MgCl_2_-6H_2_O, 0.41 mM MgSO_4_-7H_2_O, 5.33 mM KCl, 0.44 mM KH_2_PO_4_, 4.17 mM NaHCO_3_, 137.93 mM NaCl, 0.34 mM Na_2_HPO_4_, 5.56 mM D-glucose, and 20 mM HEPES) and treated with 4 μM Fluo-8 AM (AAT Bioquest, Sunnyvale, CA, USA) for 1 h. After free Fluo-8AM was thoroughly washed away with HHBS buffer, the calcium-binding activity of Fluo-8AM was detected by fluorescence spectroscopy (excitation at 490 nm and emission at 525 nm).

### 4.6. Purification of GST-Tagged PKCε

The *PKCε* gene was cloned into a glutathione-S-transferase (GST) fusion vector (pGEX-6P-1). The fusion plasmid DNA was transformed into *Escherichia coli* BL21. The transformed *E. coli* strains were grown as previously described [30]. The cells were harvested in STE buffer (150 mM NaCl, 50 mM Tris-HCl, 1 mM EDTA, 1 mM PMSF, 2 mM DTT, and 0.1% SDS) and lysed by treatment with 1 mg/mL of lysozyme and sonication. After the cell debris was removed by centrifugation, the supernatant was conjugated with GST-affinity beads for 16 h. The bead-bound PKCε-GST was eluted in an elution buffer (5 mM reduced glutathione, Tris-HCl pH 8.0). Purified PKCε-GST protein was resolved by 10% SDS-PAGE and detected by both Coomassie Brilliant Blue staining and Western blotting with an anti-PKCε antibody.

### 4.7. PKCε-DAG Binding Assay

The fluorescence spectra of the PKCε recombinant protein was obtained using a Shimadzu RF-5301PC spectrofluorometer (Kyoto, Japan). The protein samples were excited at 280 nm, and emission spectra were obtained from 300 to 390 nm. In the binding assays with PKCε, 0.89 μM protein was placed in a cuvette and a small volume of concentrated DAG solution was added to the cuvette in a stepwise manner. The fluorescence spectrum of the mixture was obtained, and the average wavelength (<λ>) was calculated according to Equation (1),
(1)$$<\lambda >=\sum F_{i}\lambda_{i}/\sum F_{i}$$
where *F_i_* represents the fluorescence intensity at a specific wavelength, *λ_i_*.

A nonlinear fitting procedure using the following equation (Equation (2)) for the binding of protein (PKCε) to DAG (L) was used to return the dissociation constant (*K_d_*),
(2)$$<\lambda >=\left({<\lambda >}_{max}\times \left[L\right]\right)/\left(K_{d}+\left[L\right]\right)$$
where <*λ*>*_max_* is the <*λ*> value at DAG saturation. The data were fitted using GraphPad Prism 6.0 (GraphPad Software, San Diego, CA, USA).

### 4.8. Fluorescence Imaging

Confluent BAECs cultured on a coverslip were transfected with a vector containing a *GFP-PKCε* fusion gene. The transfected cells were fixed with cold 4% paraformaldehyde (PFA) for 10 min at room temperature. Then the cells were mounted with a mounting solution containing 4′,6-diamidino-2-phenylindole (DAPI, Vector Laboratories, Inc., Burlingame, CA, USA) and observed under a fluorescence microscope (Axioplan 2, Carl Zeiss AG, Oberkochen, Germany).

### 4.9. THP-1 Cell Adhesion to Bovine Aortic Endothelial Cells

BAECs were starved for more than 2 h and treated with or without β-CD, TNFα and L-NAME for 16 h. THP-1 cells were stained with 1 μM Calcein AM (Sigma) for 45 min, collected by centrifugation, and thoroughly washed with DMEM. Then the stained THP-1 cells were equally aliquoted into each BAEC culture dish and additionally incubated for 1 h. After unattached THP-1 cells were removed, adherent THP-1 cells were observed under a fluorescence microscope (Axioplan 2, Carl Zeiss AG, Oberkochen, Germany).

### 4.10. Quantitative Real-Time PCR Analysis

Total RNA was collected from BAECs with NucleoZOL reagent (MACHEREY-NAGEL GmbH & Co., Dueren, Germany) and used for reverse transcription with M-MLV Reverse transcriptase (Promega, Madison, WI, USA). Then the reverse transcripts were further reacted with tTOPreal qPCR 2X preMIX SYBR Green (Enzynomics Co., Daejeon, Korea) and specific primers for real-time PCR. The primers used were as follows: VCAM-1 genes: sense, 5′-GCTGAACTGTAGCCGGAAAG-3′anti-sense, 5′-AACCGACAGCTCCTTTCTGA-3′ICAM-1 genes: sense, 5′-ACCCAAGATGAGGGTCACAG-3′anti-sense, 5′-GCTGAACTGTAGCCGGAAAG-3′.

Relative fold changes were calculated using GAPDH as a control.

### 4.11. Statistics

Data are presented as the mean±SEM and evaluated through analysis of variance (ANOVA) followed by Tukey’s post hoc multiple comparison test. Data analyses were performed using GraphPad Prism 6.0 (GraphPad Prism Software Inc., San Diego, CA, USA). The differences were considered significant if *p* < 0.05.

## Figures and Tables

**Figure 1 molecules-25-03575-f001:**
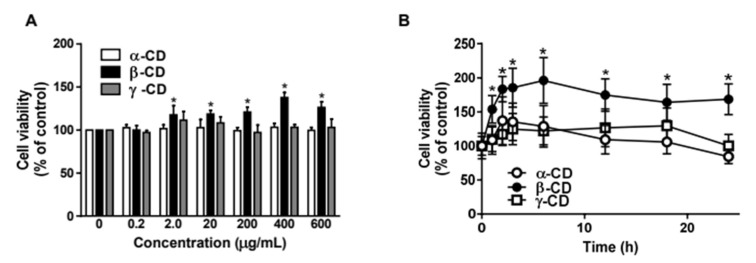
β-Cyclodextrin promotes cell proliferation in bovine aortic endothelial cells. (**A**). Bovine aortic endothelial cells (BAECs) were treated with various concentrations of α-, β-, and γ-cyclodextrin (CD) for 6 h. The BAECs were stained with Water Soluble Tetrazolium Salt (WST-1) for 1 h and then live cells were measured by the Optical Density (OD) at 450 nm. The bar graphs represent the percentage of live cells (mean ± SE, n = 3). * *p* < 0.05 vs. vehicle (one-way ANOVA and Tukey’s test). (**B**). BAECs were treated with 400 μg/mL α-, β-, and γ-CD for the time-course experiment. Then the live cells were measured as previously described in panel A. The line graphs represent the percentage of live cells (mean ± SE, n = 3). * *p* < 0.05 vs. vehicle (one-way ANOVA and Tukey’s test).

**Figure 2 molecules-25-03575-f002:**
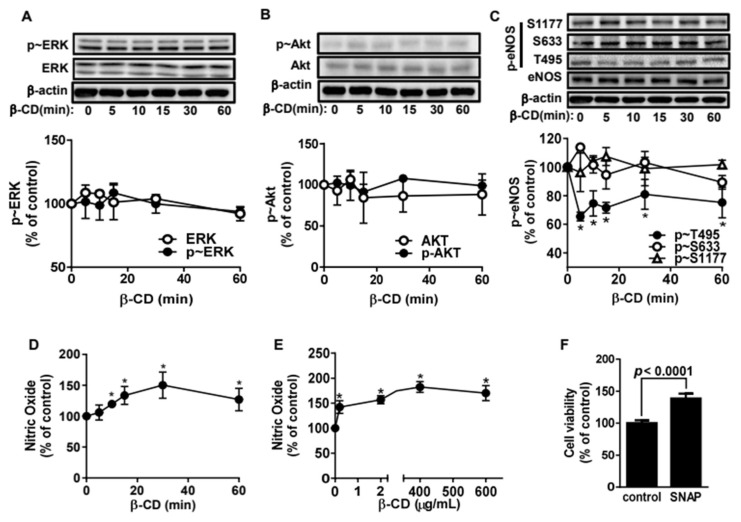
β-Cyclodextrin activates endothelial nitric oxide synthetase via dephosphorylation of threonine 495. (**A**). Bovine aortic endothelial cells (BAECs) were treated with 400 μg/mL β-CD for 0–60 min. Cell lysates were immunoblotted with the indicated anti-ERK and p-ERK antibodies. The data were quantified by densitometry and are shown in the bottom panel (mean ± SE, n = 3). (**B**). Cell lysates were immunoblotted with anti-Akt and p-Akt antibodies. The quantified data are shown in the bottom panel (mean ± SE, n = 3). (**C**). Cell lysates were immunoblotted with anti-eNOS and various p-eNOS antibodies. The quantified data are shown in the bottom panel (mean ± SE, n = 3). * *p* < 0.05 vs. vehicle (one-way ANOVA and Tukey’s test). (**D**,**E**) Nitric oxide was measured by fluorescence spectroscopy (Ex: 485 nm, Em: 530 nm) after BAECs were incubated with 2 μM DAF2-DA in the presence of 1μM A23187. The dose–response and time-course results are shown in panels D and E, respectively (mean ± SE, n = 3). * *p* < 0.05 vs. vehicle (one-way ANOVA and Tukey’s test). (**F**). Live cells were measured as described in Figure 1 after being treated with 50 μM SNAP for 12 h. The bar graph indicates the percentages of live cells (mean ± SE, n = 3). The *p* value was obtained through ANOVA followed by Tukey’s test.

**Figure 3 molecules-25-03575-f003:**
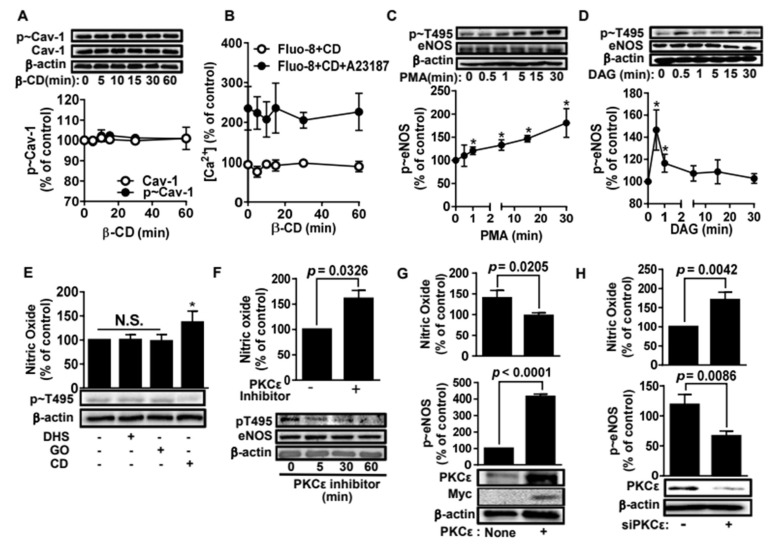
Protein kinase Cε mediates phosphorylation of endothelial nitric oxide synthetase at threonine-495 in bovine aortic endothelial cells. (**A**). Bovine aortic endothelial cells (BAECs) were incubated with 400 μg/mL β-cyclodextrin (β-CD) for the time-course experiment. Then the cell lysates were immunoblotted with anti-caveolin and p-caveolin antibodies. The quantified data are shown in the bottom panel (mean ± SE, n = 3). (**B**). BAECs were pretreated with 400 μg/mL β-CD for the indicated times and then incubated with 1 μM A23187 for 20 min. Subsequently, intracellular [Ca^2+^] was measured by fluorescence spectroscopy (Ex: 490 nm, Em: 525 nm) 30 min after incubation with 5 μM Fluo-8AM (mean ± SE, n = 3). (**C**). BAECs were incubated with 100 nM phorbol-12-myristate-13-acetate (PMA) in the time-course experiment. Cell lysates were immunoblotted with anti-eNOS and various anti-p-T495 eNOS antibodies. The data were quantified using densitometry (bottom panel, mean ± SE, n = 3). * *p* < 0.05 vs. vehicle (one-way ANOVA and Tukey’s test). (**D**). BAECs were treated with 100 μM diacylglycerol (DAG) for the indicated times. Western blots are shown in the top panels and the quantified data are shown in the bottom panel (mean ± SE, n = 3). * *p* < 0.05 vs. vehicle (One-way ANOVA and Tukey’s test). (**E**). BAECs were treated with β-CD (400 μg/mL) and PKCε inhibitors (DL-dihydrosphingosine (DHS, 10 μM) and GO6983 (GO, 10 μM)) for 30 min. Then nitric oxide (NO) and T495 phosphorylation were measured as described in Figure 2. The bar graphs represent NO levels (mean ± SE, n = 3). * *p* < 0.05 vs. vehicle (One-way ANOVA and Tukey’s test). (**F**). BAECs were treated with PKCε inhibitor peptide (20 μM) for 1 h (top panel) or the indicated times (bottom panel). Then NO levels and T-495 phosphorylation were measured as described in panel E. The *p* value was obtained through ANOVA followed by Tukey’s test. (**G**). Myc-tagged PKCε was overexpressed in BAECs. NO levels and T-495 phosphorylation were measured (mean ± SE, n = 3). The *p* value was obtained through ANOVA followed by Tukey’s test. (**H**). PKCe was knocked down using siRNA. NO levels and T-495 phosphorylation were measured (mean ± SE, n = 3). The *p* value was obtained through ANOVA followed by Tukey’s test.

**Figure 4 molecules-25-03575-f004:**
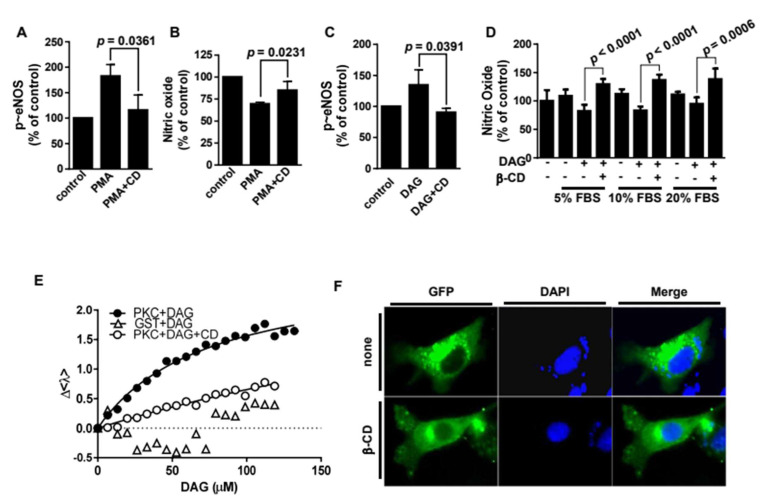
β-Cyclodextrin inhibits protein kinase Cε through molecular encapsulation of diacylglycerol. T-495 phosphorylation (**A**) and nitric oxide (NO) concentration (**B**) were measured after bovine aortic endothelial cells (BAECs) were pretreated with or without 100 nM phorbol-12-myristate-13-acetate (PMA) for 30 min and subsequently treated with β-cyclodextrin (β-CD) for 30 min. The data are shown as mean ± SE, n = 3. The *p* value was obtained through ANOVA followed by Tukey’s test. (**C**). BAECs were pretreated with or without 100 μM diacylglycerol (DAG) 30 min before β-CD treatment. Then the cell lysates were immunoblotted with an antibody against T-495 phosphorylation. The data are shown as mean ± SE, n = 3. The *p* value was obtained through ANOVA followed by Tukey’s test. (**D**). NO was measures as described in panel B. BAECs were pretreated with or without 100 μM DAG with or without β-CD in the presence of 5–20% fetal bovine serum. The data were plotted as bar graphs (means ± S.E., n = 3). The *p* value was obtained through ANOVA followed by Tukey’s test. (**E**) Purified PKCε (0.89 μM) was incubated with the indicated concentrations of diacylglycerol (DAG) with or without β-CD at a 1:1 molar ratio. The fluorescence spectra of the mixtures were obtained; the average emission wavelengths (<λ>) are shown as scatter plots. Then the data were fitted as described in the Materials and Methods section. (**F**). BAECs transfected with a control vector or a vector containing the *GFP-PKCε* gene were observed by fluorescence microscopy.

**Figure 5 molecules-25-03575-f005:**
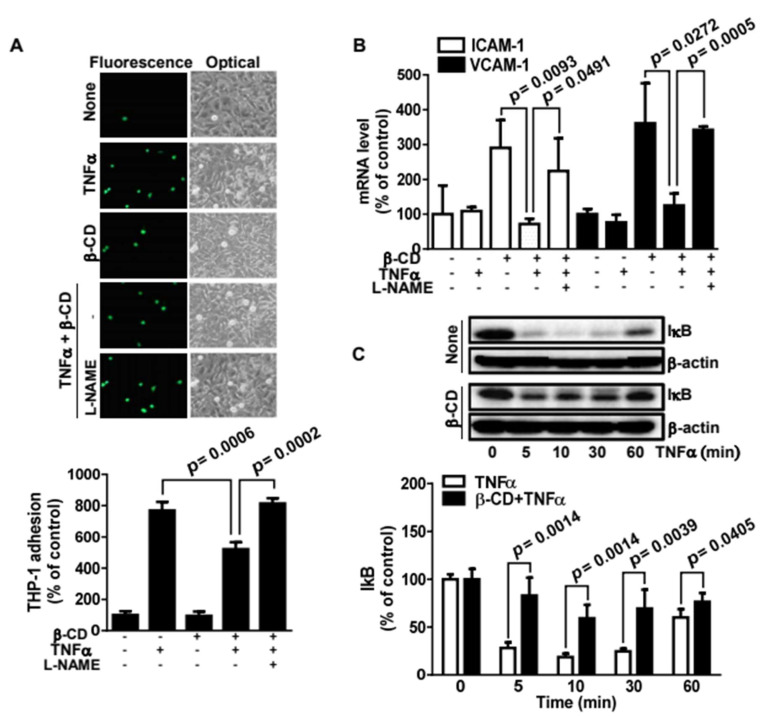
β-Cyclodextrin inhibits THP-1 cell adhesion to endothelial cells by suppressing ICAM-1 and VCAM-1 expression. (**A**). Bovine aortic endothelial cells (BAECs) were treated with or without 20 nM TNFα, 400 μg/mL β-CD, or 10 mM L-NAME for 16 h and then mixed with Calcein AM-stained THP-1 cells. After non-adherent THP-1 cells were removed, adherent cells were observed by optic or fluorescence microscopy (top panel). The number of adherent THP-1 cells is shown in the bottom panel (mean ± SE, n = 3). The *p* value was obtained through ANOVA followed by Tukey’s test. (**B**). BAECs were treated with or without 20 nM TNFα, 400 μg/mL β-CD, or 10 mM L-NAME. Then qPCR was performed using specific primers (mean ± SE, n = 4). The *p* value was obtained through ANOVA followed by Tukey’s test. (**C**). BAECs were treated with 20 nM TNFα with or without 400 μg/mL β-CD for 0–60 min. The cell lysates were immunoblotted with anti-IκB and anti-actin antibodies. Representative immunoblots are shown in the top panels and the quantified data are shown in the bottom panel (mean ± SE, n = 3). The *p* value was obtained through ANOVA followed by Tukey’s test.

**Figure 6 molecules-25-03575-f006:**
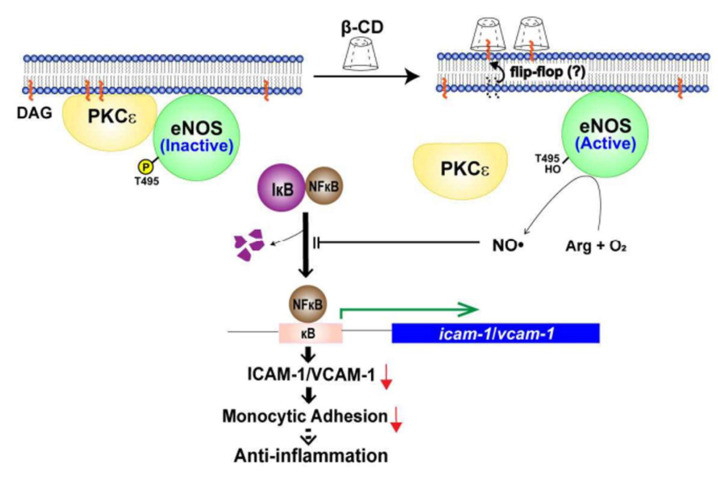
Schematic diagram of the mechanism of action of β-cyclodextrin.

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
