# Peer review of "β-Cyclodextrin Inhibits Monocytic Adhesion to Endothelial Cells through Nitric Oxide-Mediated Depletion of Cell Adhesion Molecules"

_molecules, 2020, doi:10.3390/molecules25163575_

Round 1

Reviewer 1 Report

This is a re-review of the manuscript entitled "β-Cyclodextrin inhibits monocytic adhesion to endothelial cells through nitric oxide-mediated depletion of cell adhesion molecules" submitted by Sujeong Jang, et al.  The authors revealed that β-CD inhibited the binding of PKCε to diacylglycerol (DAG) via the formation of a β-CD-DAG complex, indicating that β-CD inactivated PKCε. Furthermore, β-CD was found to control the activation of PKCε by reducing the recruitment of PKCε into the plasma membrane. In addition, the authors demonstrated that β-CD inhibits expressions of ICAM-1 and VCAM-1 by increasing NO via controlling PKCε/eNOS and suppresses THP-1 cell adhesion to endothelial cells. The manuscript is well organized and demonstrated the important role of β-CD plays in anti-inflammation. However, there are some unclear things in the manuscript. In addition, to better understand of this manuscript, the reviewer expects to add some discussion as indicated below.

  1. In the experimental section, please describe the detail experimental condition of β-CD. What kind of media was used for β-CD treatment?  Does the β-CD solution contain FBS? 
  2. The reviewer wondering whether the β-CD inhibits the binding of PKCε to DAG via the formation of a β-CD-DAG complex even in the blood, which has lots of lipids. Therefore, the reviewer would like to know the in vitro results even in the presence of FBS.
  3. The authors focused on the experiments on endothelial cells. What is the result when β-CD is treated to macrophages? Does β-CD induce iNOS from macrophages? The reviewer would like you to investigate the effect on other cells.
  4. The authors should investigate effects of β-CD on lipids rafts structure of BAECs and revealed the localization of PKC

Author Response

Reviewer #1

This is a re-review of the manuscript entitled "β-Cyclodextrin inhibits monocytic adhesion to endothelial cells through nitric oxide-mediated depletion of cell adhesion molecules" submitted by Sujeong Jang, et al.  The authors revealed that β-CD inhibited the binding of PKCε to diacylglycerol (DAG) via the formation of a β-CD-DAG complex, indicating that β-CD inactivated PKCε. Furthermore, β-CD was found to control the activation of PKCε by reducing the recruitment of PKCε into the plasma membrane. In addition, the authors demonstrated that β-CD inhibits expressions of ICAM-1 and VCAM-1 by increasing NO via controlling PKCε/eNOS and suppresses THP-1 cell adhesion to endothelial cells. The manuscript is well organized and demonstrated the important role of β-CD plays in anti-inflammation. However, there are some unclear things in the manuscript. In addition, to better understand of this manuscript, the reviewer expects to add some discussion as indicated below.

  1. In the experimental section, please describe the detail experimental condition of β-CD. What kind of media was used for β-CD treatment?  Does the β-CD solution contain FBS? 

ANSWER: As reviewer #1 suggested, we added the experimental condition of β-CD (see page 13; lines 6-7). As described in the revised manuscript, β-CD was treated in the starvation media containing 0.5% FBS.   

  1. The reviewer wondering whether the β-CD inhibits the binding of PKCε to DAG via the formation of a β-CD-DAG complex even in the blood, which has lots of lipids. Therefore, the reviewer would like to know the in vitro results even in the presence of FBS.

ANSWER: To answer the reviewer #1's concerns, we executed additional experiments to know whether β-CD inhibits the physiological function of PKCε in response to DAG in the presence of FBS (up to 20%). For this experiment, NO measurement was used as a parameter for the functional activity of PKCε associated with DAG. As shown Fig. 4D in the revised manuscript, β-CD blockaded DAG/PKCε-promoted NO in the presence of the high concentration of serum, suggesting that a β-CD-DAG complex may occur in the blood. This result was described in the revised manuscript (page 8; lines 9-11). We also tried to see the inhibitory effect of β-CD through in vitro binding assay, but FBS enormously elevated the background (noise) of fluorescence, thus not feasible.       

  1. The authors focused on the experiments on endothelial cells. What is the result when β-CD is treated to macrophages? Does β-CD induce iNOS from macrophages? The reviewer would like you to investigate the effect on other cells.

ANSWER: Firstly, we tested if β-CD induces iNOS in macrophage. By using real-time PCR, iNOS messenger was determined after THP-1 cells were treated with β-CD. As shown in below Figure, β-CD had no effect on iNOS expression. We did not discuss this result in the revised manuscript. (Figure can be seen in attached file)

Secondly, we examined whether β-CD has an effect on THP-1 cells. Interestingly, TNFα-induced adhesion of THP-1 cells to BAECs was remarkable reduced when THP-1 cells were pre-conditioned with β-CD. This result indicates that β-CD act as an anti-inhibitory agent in other cells. In the present study, we do not know the MoA of β-CD in THP-1 cells, but we discussed this result in the revised manuscript (see Supplementary Fig. 3 & text page 12; lines 14-19).    

  1. The authors should investigate effects of β-CD on lipids rafts structure of BAECs and revealed the localization of PKC

ANSWER: As reviewer #1 suggested, we investigated effects of β-CD on lipids rafts. Firstly, we executed the immunostaining using anti-caveolin-1 antibody. As shown in Supplementary Fig 1, β-CD did not alter both punctate caveolin-1 staining and PKCε localization in lipid raft fractionation (Triton-insoluble fraction). These results were described in the revised manuscript (see page 7; lines 9-10). In addition, co-localization of caveolin-1 and PKCε was evaluated by using immunostaining. Interestingly, we found that PKC is rarely co-localized with caveolin-1 (Supplementary Fig. 1B & Supplementary Fig. 4). These data was discussed in the revised manuscript (see page 9; lines 1-2).      

Reviewer 2 Report

The introduction must be improved. The authors must write the biological useful of Cyclodextrins (CDs) in popular medicine or in in vivo studies that can shown the pharmacologies properties, as an anti-inflammatory agents

The figure are not good

The statistical significances are not so important

The stastistical analysis results must be described in the results

It is important to separate results from discussion section

The english style must be improved

Author Response

Reviewer #2

  1. The introduction must be improved. The authors must write the biological useful of Cyclodextrins (CDs) in popular medicine or in in vivo studies that can shown the pharmacologies properties, as an anti-inflammatory agents

ANSWER: As reviewer #1 suggested, the Introduction section was supplemented by addition of the context regarding the bio-applications of CDs in medicine or pharmacology as an anti-inflammatory agents in the revised manuscript (see page 3; lines 19-22 & page 4; lines 1-5).

  1. The figure are not good

ANSWER: The figures were re-drawn in the revised manuscript as reviewer #2 suggested 

  1. The statistical significances are not so important

ANSWER: Statistics was re-written in the Materials and Methods section (see page 18; lines 5-8)

  1. The statistical analysis results must be described in the results

ANSWER: Statistical results were indicated in each datum point in the revised manuscript (see Figure 1-5 Legends in the revised manuscript).

  1. It is important to separate results from discussion section

ANSWER: We separated results from discussion section in the revised manuscript.

  1. The english style must be improved

ANSWER: English was edited by English editing company (see the certificate in the attached file)

Round 2

Reviewer 1 Report

The authors revised the manuscript as much as they can. The reviewer understand well the content of this manuscript.  

Reviewer 2 Report

The authors answered all the questions raised by Referee. All the modifications are good for publication